# Simultaneous Monitoring of Mutation and Chimerism Using Next-Generation Sequencing in Myelodysplastic Syndrome

**DOI:** 10.3390/jcm8122077

**Published:** 2019-11-28

**Authors:** Jong-Mi Lee, Yoo-Jin Kim, Sung-Soo Park, Eunhee Han, Myungshin Kim, Yonggoo Kim

**Affiliations:** 1Department of Laboratory Medicine, College of Medicine, The Catholic University of Korea, Seoul 06591, Korea; jongmilee86@gmail.com (J.-M.L.); haniyes@catholic.ac.kr (E.H.); 2Catholic Genetic Laboratory Center, Seoul St. Mary’s Hospital, College of Medicine, The Catholic University of Korea, Seoul 06591, Korea; 3Department of Hematology, Seoul St. Mary’s Hematology Hospital, College of Medicine, The Catholic University of Korea, Seoul 06591, Korea; yoojink@catholic.ac.kr (Y.-J.K.);

**Keywords:** chimerism, next-generation sequencing (NGS), short tandem repeat (STR), minimal residual disease (MRD), allogeneic hematopoietic stem cell transplantation (allo-HSCT), myelodysplastic syndrome (MDS)

## Abstract

Monitoring minimal residual disease (MRD) provides important information during treatment of hematologic malignancies. Chimerism analysis also provides key information after allogeneic hematopoietic stem cell transplantation (allo-HSCT). Recent advances in next-generation sequencing (NGS) have enabled identification of various mutations and quantification of mutant allele burden. In this study, we developed a new analytic algorithm to monitor chimerism applicable to NGS multi-gene panel in use to identify mutations of myelodysplastic syndrome (MDS). We enrolled patients who were diagnosed with MDS and received allo-HSCT and their corresponding donors. Monitoring MRD by NGS assay was performed using 53 DNA samples by calculating mutant allele burden after treatment. For monitoring chimerism by NGS, we selected 121 single nucleotide polymorphisms (SNPs) after careful stepwise evaluation and calculated average donor allele burden. Data obtained from NGS were compared with bone marrow findings, chromosome analysis and short tandem repeat (STR)-based chimerism. SNP-based NGS chimerism analysis was accurate and even superior to conventional STR method by overcoming the various technical limitations of STR. In addition, simultaneous monitoring of mutation and chimerism using NGS could implement comprehensive pre- and post-HSCT monitoring of various clinical conditions such as complete donor chimerism, persistent mixed chimerism, early relapse, and even donor cell-derived diseases.

## 1. Introduction

Monitoring minimal residual disease (MRD) provides important information during treatment of hematologic malignancies. Various techniques can be used to analyze genetic alterations in hematologic malignancies; therefore, selecting the appropriate technique for each patient is the initial step of MRD monitoring. Reverse transcription quantitative PCR can be used to measure fusion transcripts, but it cannot be applied in patients without gene fusion. Recurrent mutations such as those in the *NPM1* gene are also MRD markers [1]. However, the majority of mutations vary by patient, so analyzing individual mutations via quantitative PCR in a patient-specific setting remains challenging. Recent advances in next-generation sequencing (NGS) have enabled identification of various mutations and quantification of allele burden in acute myeloid leukemia (AML). Thus, using NGS for both identification and MRD monitoring is becoming increasingly feasible [2,3]. Myelodysplastic syndrome (MDS) is a genetically heterogeneous disorder similar to AML, and various associated somatic mutations have been identified in more than 50 genes [4]. Because recurrent gene fusions or mutations are rare in MDS, multi-gene targeting NGS is a suitable technique for MRD monitoring [5]. In addition, monitoring of chimerism provides key information after allogeneic hematopoietic stem cell transplantation (allo-HSCT). Multiplex PCR amplification of short tandem repeat (STR) markers is a commonly used technique because of its high applicability and accuracy with 1–5% sensitivity [6]. Real-time quantitative PCR of insertion/deletion markers showed higher sensitivity with a 0.1% threshold compared with that of STR analysis [7]. Recently, a custom NGS chimerism panel using single-nucleotide polymorphism (SNP) markers has been applied to chimerism and showed a high sensitivity of 0.5–1%, which was higher than that of STR analysis [8,9]. Hence, the optimal method for monitoring MRD in MDS both pre- and post- HSCT may require more than one technology [10]. In this study, we developed a new analytic algorithm using a clinically used NGS myeloid panel that simultaneously monitors mutation and chimerism in MDS. Chimerism was calculated from allele burden of SNPs included in the NGS panel after careful stepwise evaluation.

## 2. Experimental Section

### 2.1. Ethics Statement

This study was performed in accordance with the Declaration of Helsinki and approved by the Institutional Review Board at Seoul St. Mary’s Hospital (KC16SISI0395). The IRB waived the requirement to obtain informed consent from participants, because this is the retrospective study of the clinical cases involving minimal risk to the patients.

### 2.2. Subject and DNA Isolation

We enrolled 14 patients who were diagnosed with MDS and received allo-HSCT at Seoul St. Mary’s Hematology Hospital and their corresponding donors. To investigate the most effective approach for the simultaneous detection of the mutation and chimersim, we have included all possible conditions after allo-HSCT including complete donor chimerism, mixed chimerism and donor cell-derived MDS. Medical records of the patients were carefully reviewed including bone marrow pathologic findings and chromosomal analysis. Fifty three samples were obtained from donors (*n* = 14), patients at the time of diagnosis (*n* = 14) and after HSCT (*n* = 25). DNA was extracted from peripheral blood or bone marrow (BM) aspirates using a column-based DNA isolation technique (QIAamp DNA Blood mini kit, QIAGEN, Hilden, Germany). DNA concentration and purity was checked by ND-1000 spectrophotometry (Nanodrop Technologies, Wilmington, DE, USA). 

### 2.3. STR Analysis

STR analysis was performed using AmpFlSTR Identifier PCR Amplification (Applied Biosystems, Warrington, UK) as previously reported [11]. Briefly, 16 STR markers were amplified; D8S1179 at chromosome 8, D21S11 at 21q11.2–q21, D7S820 at 7q11.2–22, CSF1PO at 5q33.3–34, D3S1358 at 3p, TH01 at 11p15.5, D13S317 at 13q22–31, D16S539 at 16q24-qter, D2S1338 at 2q35–37.1, D19S433 at 19q12–13.1, vWA at 12p12-pter, TPOX at 2p23-2per, D18S51 at 18q21.3, D5S818 at 5q21–31, and FGA at 4q28, the amelogenin locus at X (p22.1–22.3) and Y (p11.2) chromosomes. PCR was performed using a C1000 Touch™ Thermal Cycler (Bio-Rad laboratories Inc., Hercules, CA, USA). Amplified PCR products were analyzed by capillary electrophoresis using an ABI 3130xl genetic analyzer (Applied Biosystems, Foster City, CA, USA). GeneMapper ID Software Version 4.1 (Applied Biosystems, Foster City, CA, USA) was used for automated genotyping and quantification of peak areas.

### 2.4. Customised NGS Panel Analysis 

NGS was performed using a customized myeloid panel containing 87 genes frequently mutated in patients with MDS and myeloproliferative neoplasia (Appendix A). Target capture sequencing was performed using a customized target kit (3039061, Agilent Technologies, Santa Clara, CA, USA) according to the manufacturer’s instructions. DNA libraries were constructed according to the protocol, and the customized target kit was performed using an Illumina HiSeq4000 platform to generate 101 bp paired-end reads. We used cutadapt [12] and sickle (https://github.com/najoshi/sickle, accessed on 29 October 2015) for removing adapter sequences and low-quality sequence reads. Burrows-Wheeler aligner [13] was used to align the sequencing reads onto the human reference genome (hg19). We used a Genome Analysis ToolKit (GATK) [14] for local realignment, score recalibration, and filtering of sequence data. Picard (https://github.com/broadinstitute/picard, accessed on 18 May 2015) and Samtools [15] were also used for basic processing and management of the sequencing data and generated mpileup file. VarScan v.2.3.9. (http://varscan.sourceforge.net/, accessed on 16 September 2015) was used to call variants. The donor chimerism and mutant allele burden (MAB) were calculated using reads counts for the four bases (A,C,G,T) on target sites retrieved by Samtools mpileup and sequenza [16]. Figure 1 illustrates the analytical work flow. 

### 2.5. Minimal Residual Disease Monitoring 

Annotated variants were further classified into four tiers according to the Standards and Guidelines by the Association for Molecular Pathology (AMP) [17]. All the variants with minor allele frequency >0.01 were filtered out based on the Exome Aggregation Consortium (ExAC, http://exac.broadinstitute.org/) and genome aggregation database (gnomAD, https://gnomad.broadinstitute.org/), as well as an ethnic-specific Korean Variant Archive (KOVA, http://kobic.re.kr /kova/). The variants, reported more than three times in the hematopoietic tissues in the Catalogue Of Somatic Mutations In Cancer database (COSMIC, https://cancer.sanger.ac.uk/cosmic) were included. In addition, nonsense, frameshift, or splice site variants were included when the known mechanism of the mutation was loss-of function. All of the variants were manually verified using the Integrative Genomic Viewer. We finally selected the leukemia-associated mutations from detected variants as MRD markers. Limit of blank was determined by mean % background error (BE) + 3 standard deviations (SD) loci (Table 1) [18].

### 2.6. SNP-Based NGS Chimerism Analysis 

For NGS chimerism analysis, we reviewed all the identified SNPs as the frequency of heterozygosity in the general Korean population (the Korean reference genome database investigated 1722 Korean individuals). The homozygous and heterozygous alleles were determined by base count frequency of 90–100% and 45–60% [8], respectively. We examined 153 SNPs with a frequency of heterozygosity ranging from 0.2 to 0.8 in the Korean database and selected optimal SNPs for NGS chimerism analysis based on the following criteria: (1) >500 mean read depth, (2) ≤0.2% BE, and (3) <10% measurement error of heterozygous alleles (ME, difference of read count between reference and alternative alleles) [19]. Finally, 121 SNPs were selected for NGS chimerism analysis (Appendix A), and the average read depth, %BE, and %ME of selected SNPs were 1398.3 ± 538.9, 0.082% ± 0.035%, and 3.31% ± 2.27%, respectively. Donor allele burden of each SNP was calculated according to the following formula; A/(A+a) [Donor(AA) and Recipient(aa)], a/(A+a) [Donor(aa) and Recipient(AA)], 1-2x{a/(A+a)} [Donor(AA) and Recipient(Aa)], 1-2x{A/(A+a)} [Donor(aa) and Recipient(Aa)]. Donor chimerism was defined as the average donor allele burden [19].

## 3. Results

### 3.1. Analytical Performance of NGS Donor Chimerism 

Mean on-target read depth ranged from 1406.81× to 2052.61×, and the 50× coverage rates ranged from 99.34% to 99.75%. We did not detect any donor carrying a clonal hematopoiesis-related mutation [20]. The number of informative SNPs was variable (median 25.5, range 9–41) and higher than that of STR (median 10.5, range 5–14). Donor chimerism from NGS analysis highly correlated with results from STR analysis, with *r*^2^ = 0.994 (*p* < 0.0001) by least-squares analysis (Figure 2a), and all data points were within 3 SDs of the mean % difference on Blant-Altman plot (Figure 2b). In addition, we calculated positive (PPV) and negative predictive value (NPV) for our test to diagnose complete donor chimerism with three cut-off values including 99%, 95% and 90% [21,22,23]. The overall agreements between the NGS and STR analyses were 100% when we applied the 95% and 90% cut-off values. When applying 99% cut-off value, NGS analysis showed 100% sensitivity, 91.7% specificity, 92.3% PPV and 100% NPV. Because our laboratory uses 95% cut-off value for complete chimerism, current agreement between two methods was 100% in qualitative comparison.

### 3.2. Clinical Usefulness of Simultaneous Monitoring

Nine patients (1–9) showed complete donor chimerism (STR 99.29 ± 0.76%, NGS 99.60 ± 0.58%) with 0.03 ± 0.05% MAB (Figure 3). 

Patient 10 maintained mixed chimerism with normal hematologic findings. Among the 14 patients, three patients (11–13) relapsed after allo-HSCT. We detected increased MAB and decreased donor chimerism at relapse (patient 11, Figure 4a). Patient 12 showed mixed chimerism at the first year follow-up and persistent *SF3B1* mutation at a low level (0.44 ± 0.16, Figure 4b). Patient 13 showed mixed chimerism until 7 months after allo-HSCT. *NRAS* mutation was observed at diagnosis, but it disappeared after allo-HSCT. He relapsed to AML at 9 months after allo-HSCT with decreased donor chimerism. Notably, a new cytogenetic aberration was obtained without reappearing *NRAS* mutation (Figure 4c). Patient 14 verified the effect of the analysis algorithm. She was treated for AML and received allo-HSCT 11 years ago. During regular follow-up, pancytopenia occurred, and BM examination was performed for hematopathologic evaluation and chimerism analysis, revealing 3-lineage dysplasia and increased blasts (5%). Notably, complete donor chimerism (99.8%) along with *PHF6* mutation was identified by NGS, which suggested that the mutation originated from donor cells (Figure 4d). This impression was further supported by cytogenetic analysis as //46,XY,+1,der(1;7)(q10;q10)[5]/46,XY[5], which demonstrated donor-cell derived MDS accompanied by 1q gain and 7q loss. Patient characteristics and the results of donor chimerism and mutant burden are described in Table 2.

## 4. Discussion

In hematologic malignancies, risk stratification and clinical decision are made based on monitoring disease after treatment. Among available technologies for MRD monitoring, multi-parameter flow cytometry and measuring mutant burden by quantitative PCR are sensitive and specific. But they are restricted for patients with particular aberrant expression or molecular markers and do not detect evolving clonal aberrations [24,25,26]. NGS has great potential for MRD monitoring because it has ability to determine multiple mutations simultaneously with clonal burden [10,27]. After allo-HSCT, patients are monitored by chimerism analysis and routine hematology test at regular intervals. Chimerism analysis is highly applicable to most patients after allo-HSCT, but commonly used STR assay is less sensitive and specific because leukemic cells are not directly targeted. CD34-positive cell sorted chimerism may overcome some part of the limitations, but CD34 expression is variable and cell sorting process is laborious. Another important consideration is multiple meaning of mixed chimerism. The mixed chimerism can have various clinical implications including disease relapse, graft failure or rejection. However, previous studies showed that mixed chimerism may remain stable over long time and be compatible with prolonged remission [28]. Moreover, there are increasing states of mixed chimerism, especially after reduced intensity conditioning regimens and after T-cell depletion [28]. Therefore, results from chimerism analysis should be comprehensively interpreted and it is desirable to combine the different methods to strengthen the strength and make up for the weakness of them. 

In this study, we developed a new analytic algorithm to monitor chimerism applicable to NGS multi-gene panel in use to identify mutations of MDS through careful stepwise evaluation. This is the first attempt to analyze both chimerism and mutation simultaneously, especially through a novel but simple analytical algorithm. Through this algorithm, we could implement comprehensive pre- and post-HSCT monitoring of various clinical conditions such as complete donor chimerism, persistent mixed chimerism, early relapse, and donor cell-derived MDS. Patients with complete donor chimerism showed MAB less than threshold. Among patient with mixed chimerism, three relapsed showed increased MAB and decreased donor chimerism. The other patient relapsed to AML with decreased donor chimerism without reappearing initially detected *NRAS* mutation. This result was in line with a previous study that showed some mutations to be effectively eliminated through HSCT [24]. Another patient maintained stable mixed chimerism. These findings indicated that the individual disease process is effectively demonstrated through simultaneous analysis. Notably, we successfully detected a donor-cell derived MDS patient showing complete donor chimerism and evolving clonal mutation. Donor cell-derived hematologic malignancy is an infrequent complication after all-HSCT. Its diagnosis can be delayed until blasts emerge in peripheral blood because STR shows complete donor chimerism. Previous report reviewed literatures and identified more than 70 cases of donor cell-derived leukemia (DCL). Time between allo-HSCT and occurrence of DCL was various (median 30 months, range 1-279) [29]. Although DCL can be diagnosed by chmerism analysis, it is more important to detect the genetic landscape and factors which contribute to DLC such as germline mutations [30]. It can help us to understand pathogenesis of DCL and to make better therapeutic plans. The other important interesting condition is pre-existing clonal hematopoiesis in donor [20]. We did not detect clonal hematopoeisis –related mutations in donor, however, it is necessary to monitor if the mutations were detected in donor cells. Our simultaneous analysis using NGS is very useful to evaluate these particular conditions after allo-HSCT.

This method is very convenient and cost-effective because it can be applied to any NGS panel for hematologic malignancies after selection and evaluation of informative SNPs. The turn-around time of the simultaneous analysis was not longer than that of mutation analysis using NGS (about 4 days) because SNP-based chimerism calculation can be performed simply with minimal time (less than 30 min). The number of informative SNPs was enough to analyze chimerism and the NGS chimerism was concordant with STR assay result. The overall agreements between the NGS and STR analyses to diagnose complete chimerism were excellent. We postulated that SNP-based NGS chimerism analysis would be superior to STR assay because NGS overcomes the various technical limitations of STR assay, such as stutter peak, peak height imbalance, non-template adenine addition, dye interference, voltage spikes, and allele dropouts [9,31]. Although we tried to minimize the bias of conventional NGS methodology through excluding SNPs with low read depth, high background error rate and allelic imbalance, there are still technical limitations such as repetitive amplification of the same reads. This can be overcome by barcoded error-corrected sequencing methods such as single molecule molecular inversion probes (smMIPs) [32,33]. Development of improved error-corrected sequencing method to increase the sensitivity and specificity of NGS technology coupled with automated calculation system would potentiate the usefulness of this novel analytic algorithm. And it is worthy to validate the analytical and clinical performance of this improved method through a large number of cases in a prospective manner. 

## 5. Conclusions

In this study, we developed a new analytic algorithm using a clinically used NGS myeloid panel that simultaneously monitors mutation and chimerism. This method is applicable to any NGS panels and allows chimerism analysis from allele burdens of SNPs included in the NGS panel. This approach showed excellent performance and provided useful information to understand various clinical status after allo-HSCT.

## Figures and Tables

**Figure 1 jcm-08-02077-f001:**
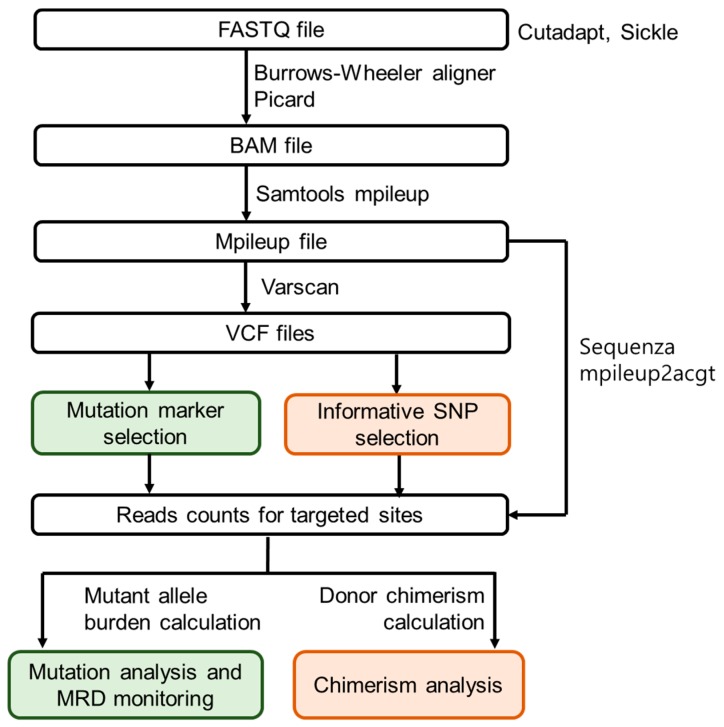
Algorithm for simultaneous analysis of chimerism and mutation. The A,C,G,T counts at the positions of mutation markers and informative single nucleotide polymorphisms (SNPs) were extracted from Mpileup file using Sequenza and mpileup2acgt. BAM: Binary version of Sequence Alignment Map, VCF: Variant Calling Format, SNP: Single Nucleotide Polymorphism, MRD: Minimal Residual Disease.

**Figure 2 jcm-08-02077-f002:**
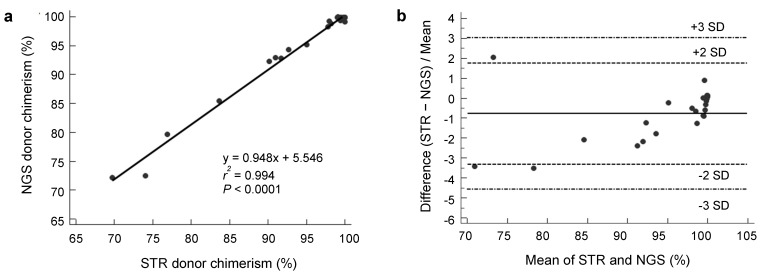
(**a**) Scatter plot of donor chimerism values obtained by next-generation sequencing (NGS) and STR shows excellent correlation. (**b**) Blant-Altman plot of the two chimerism assays.

**Figure 3 jcm-08-02077-f003:**
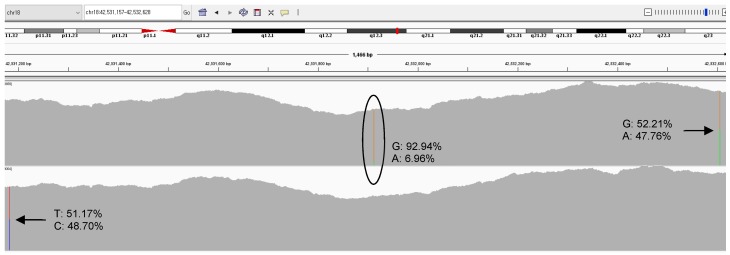
Representative integrative genomic viewer at *SETBP1* region of pre- (upper chart) and post- hematopoietic stem cell transplantation (HSCT) (lower chart) samples from patient 1. Somatic mutation (6.96%, green area in circle) disappeared and single nucleotide polymorphisms (arrows) changed after HSCT.

**Figure 4 jcm-08-02077-f004:**
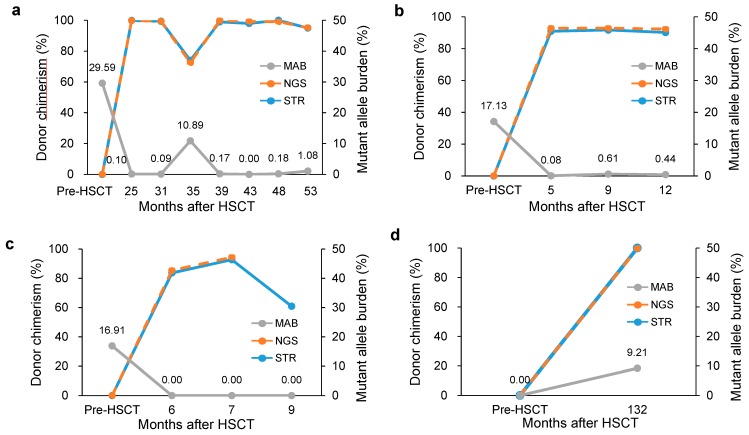
Kinetics of engraftment monitoring using NGS, STR and mutant allele burden (MAB) on longitudinal follow-up samples before and after hematopoeitc stem cell transplantation (HSCT) of patient 11 (**a**), 12 (**b**), 13 (**c**) and 14 (**d**).

**Table 1 jcm-08-02077-t001:** Determination of limit of blank for monitoring minimal residual disease.

Mutation	Location	Read Depth	%BE	%SD
*NRAS*:c.35G>A	Chr1:115258747	2048.7 ± 416.2	0.026	0.041
*SF3B1*:c.2098A>G	Chr2:198266834	1930.5 ± 365.2	0.018	0.031
*SETBP1*:c.2608G>A	Chr18:42531913	1598.5 ± 313.4	0.041	0.064
*U2AF1*:c.101C>A,T	Ch21:44524456	1410.7 ± 351.4	0.026	0.055
*PHF6*:c.820C>T	ChrX:133549136	1184.0 ± 455.8	0.023	0.052

BE, background error; Chr, chromosome; Read depth described as mean ± SD.

**Table 2 jcm-08-02077-t002:** Characteristics of patients with donor chimerism and mutant burden.

Case Type	No.	Gender/Age	Type of HSCT	Status	Sample	F/U Months	BM Findings Cellularity (Blasts)	Chromosome Analysis	%Donor STR	No. IRA STR	%Donor NGS	No. IRA NGS	MAB (%) ^†^	Mutation
**Complete donor chimerism**	1	M/61	Unrelated matched - PBSCT	Pre	PB		RCMD *	46,XY[20] *					**22.40**	*U2AF1*:c.101C>T
													**6.60**	*SETBP1*:c.2608G>A
				Post 1	PB	30	NA	NA	99.4	14	99.94	41	0.00	*U2AF1*:c.101C>T
													0.00	*SETBP1*:c.2608G>A
	2	F/39	Family mismatched -PBSCT	Pre	PB		Hypo-MDS *	46,XX,t(2;3)(p23;q29)[14]/ 46,XX[6] *					**10.33**	*SF3B1*:c.2098A>G
				Post 1	BM	25	30% (1%)	46,XX[20]	100	8	99.87	24	0.00	
	3	M/29	Family mismatched -PBSCT	Pre	PB		MDS EB-1 *	46,XY[20] *					**31.78**	*U2AF1*:c.101C>T
				Post 1	PB	46	NA	NA	99.1	11	99.97	31	0.00	
	4	F/29	Sibling matched -PBSCT	Pre	PB		RCMD *	47,XX,+8,del(11)(q21)[8]/ 46,XX[12] *					**37.64**	*U2AF1*:c.101C>A
				Post 1	PB	54	NA	NA	99.9	5	99.91	17	0.15	
	5	M/45	Sibling matched -PBSCT	Pre	PB		MDS EB-2 *	46,XY[15] *					**26.22**	*U2AF1*:c.101C>T
				Post 1	PB	37	NA	NA	100	6	99.93	27	0.00	
	6	F/26	Sibling matched -PBSCT	Pre	PB		RCMD *	47,XX,+8[3]/46,XX[17] *					**6.88**	*U2AF1*:c.101C>A
				Post 1	PB	32	NA	NA	97.8	14	98.28	30	0.10	
	7	M/58	Family mismatched -PBSCT	Pre	PB		RCMD *	46,XY[20] *					**46.74**	*U2AF1*:c.101C>T
													**3.69**	*NRAS*:c.35G>A
				Post 1	BM	10	50% (<1%)	46,XY[20]	98.2	13	98.81	28	0.00	*U2AF1*:c.101C>T
													0.00	*NRAS*:c.35G>A
	8	M/24	Family mismatched -PBSCT	Pre	PB		RCMD *	47,XY,+8[20] *					**7.10**	*U2AF1*:c.101C>A
				Post 1	PB	37	NA	NA	99.6	7	99.87	11	0.08	
	9	F/31	Sibling matched -PBSCT	Pre	PB		MDS EB-1 *	46,XX[20] *					-	None
				Post 1	PB	30	NA	NA	99.7	10	99.86	20	-	
**Mixed chimerism**	10	M/3	Sibling matched - BMT	Pre	PB		RCC *	46,XY[20] *					-	None
				Post 1	PB	1	20% (1%) *	46,XY[2]//46,XX[28] *	76.9	9	80.56	9	-	
				Post 2	PB	3	80% (<1%) *	46,XY[8]//46,XX[12] *	69.7	6	72.00	10	-	
	11	F/59	Unrelated matched - PBSCT	Pre	PB		RCMD *	46,XX[20] *					**29.59**	*U2AF1*:c.101C>T
				Post 1	PB	25	NA	NA	99.9	14	99.88	37	0.10	
				Post 2	PB	31	NA	NA	99.4	14	99.39	37	0.09	
				Post 3	BM	35	70% (18%)	47,XX,+21[9]//46,XY[11]	74.0	14	72.49	28	**10.89**	
				Post 4	PB	39	NA	NA	99.0	14	99.85	37	0.17	
				Post 5	BM	43	30% (5%)	//46,XY[30]	98.0	14	99.23	37	0.00	
				Post 6	PB	48	NA	NA	100	14	99.09	37	0.18	
				Post 7	PB	53	5% (18%) *	47,XX,+21[10]//46,XY[10] *	95.0	14	95.23	37	**1.08**	
	12	M/54	Sibling matched - BMT	Pre	PB		RARS *	46,XY[12] *					**17.13**	*SF3B1*:c.2098A>G
				Post 1	PB	5	15% (<1%) *	46,XY[20] *	90.9	7	92.91	19	0.08	
				Post 2	PB	9	30% (<1%) *	46,XY[20] *	91.7	8	92.84	19	**0.61**	
				Post 3	PB	12	10% (3%) *	46,XY[10] *	90.2	8	92.35	19	**0.44**	
	13	M/52	Sibling matched - BMT	Pre	PB		MDS EB-2 *	46,XY[20] *					**16.91**	*NRAS*:c.35G>A
				Post 1	PB	6	30% (<1%) *	46,XY[20] *	83.7	9	85.44	9	0.00	
				Post 2	BM	7	15% (1%)	46,XY[20]	92.7	8	94.37	9	0.00	
				Post 3	BM	9	40% (5%)	46,XY,t(1;21)(p36.3;q11.2)[19]/ 46,XY[1]	60.9	9	NA	NA	0.00	
**Donor-cell derived MDS**	14	F/59	Sibling matched - BMT	Pre	PB		MDS EB-1 *	NA *					0.00	*PHF6*:c.820C>T
				Post 1	BM	132	15% (5%)	//46,XY,+1,der(1;7)(q10;q10)[5]/46,XY[5]	100	5	99.80	13	**9.21**	

* Data within 3 months of chimerism analysis. ^†^ Values above threshold of minimal residual disease were written in bold. HSCT, hematopoietic stem cell transplantation; PBSCT peripheral blood stem cell transplantation; BMT, bone marrow transplantation; RCMD refractory cytopenia with multilineage dysplasia; NA, not available; Hypo-MDS, hypocellular myelodysplastic syndrome; EB, excess blasts; RCC, refractory cytopenia of childhood; RARS, refractory anemia with ring sideroblasts; F/U months, months after HSCT; %Donor, Donor chimerism; IRA, informative recipient alleles; MAB, mutant allele burden.

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
