# Peer review of "Simultaneous Monitoring of Mutation and Chimerism Using Next-Generation Sequencing in Myelodysplastic Syndrome"

_jcm, 2019, doi:10.3390/jcm8122077_

Round 1

Reviewer 1 Report

Lee et al. reported an exciting result of the study with a novel idea of simultaneous monitoring of mutation and chimerism using NGS in MDS patients. The current standard for chimerism monitoring test is STR or VNTR based analysis, which includes multiple STR markers. The main idea is to analyze NGS-based sequencing data, which cover not only hotspot for somatic mutation but also SNPs, thus to calculate the proportion of DNAs from a donor vs that from a recipient. However, there is some weakness in the paper, as pointed out below. This paper is worth to be published but will require major revision.

Major comments

Donor samples seemed sequenced, but it was not clearly described in the paper. Please clarify that with its sequencing depth. Also, please mention that if there is any case of donor carrying a clonal hematopoiesis-related mutation. Variant calling algorithm. Please provide details of the variant calling algorithm for chimerism measurement. Did you use integrative genomic viewer for chimerism? If it is the case, it is not an automated procedure that increases the chance of human error for chimerism measurement. Any comment? Another pitfall of the conventional NGS methodology is the repetitive amplification of the same reads, which will affect the calculation of chimerism. In order to overcome this limitation, there is a lot of technology development incorporating bar-coded error-corrected sequencing methods such as smMIP probe development. I wonder if authors have a plan to apply this methodology to improve variant calling for allele frequency. At least it has to be included in the discussion. A very subtle change of chimerism would also affect significantly decision making for the clinical management of the case. Thus more accurate methodology for bar-coded error-corrected sequencing is needed for this purpose as well as more solid algorithm is required for variant calling. If your goal of the study is to demonstrate that NGS based chimerism measurement is as useful as STR-based chimerism test, not only the correlation analysis between the two methods (as shown in Figure 1A) but also you need to calculate positive and negative predictive value for your test as well as sensitivity/specificity. For section 3.2 (page 4, line 148), there is result from the 14 patients. Are there any predetermined criteria for case selection? For example, did you intend to include some of full donor chimerism and some of mixed chimerism as well as a case of donor cell-derived leukemia? I was not sure the last case of donor cell-derived leukemia is a relevant case for this approach. If yes, please highlight what is the benefit of using simultaneous monitoring of chimerism and mutation based on NGS based method.

Minor comments

Table 2 is labelled as Table 1 (line 176). Please add a column to label the case as mixed chimerism or donor-cell derived leukemia etc. That would help readers to have a better understanding of Table 2. Figure 3 does not include the definition of abbreviations used in the figure. Turn-around time of reporting has never been described in the paper. Any comment?

Author Response

Response to Reviewer 1 Comments

Lee et al. reported an exciting result of the study with a novel idea of simultaneous monitoring of mutation and chimerism using NGS in MDS patients. The current standard for chimerism monitoring test is STR or VNTR based analysis, which includes multiple STR markers. The main idea is to analyze NGS-based sequencing data, which cover not only hotspot for somatic mutation but also SNPs, thus to calculate the proportion of DNAs from a donor vs that from a recipient. However, there is some weakness in the paper, as pointed out below. This paper is worth to be published but will require major revision.

Thank you very much for your sincere and warm acknowledgment about our manuscript. In the following sections, you will find our responses to each of your points and suggestions.

Point 1: Donor samples seemed sequenced, but it was not clearly described in the paper. Please clarify that with its sequencing depth. Also, please mention that if there is any case of donor carrying a clonal hematopoiesis-related mutation.

 Response 1: Fifty three samples were obtained from donors (n = 14), patients at the time of diagnosis (n = 14) and patients after hematopoietic stem cell transplantation (n = 25). We have commented the source of samples in revised manuscript. We also have clarified the sequencing depth including ranges of mean read-depth and 50x coverage rates of the total 53 samples in the results section.

Thank you for pointing a clonal hematopoiesis in donor. We carefully reanalyzed donors’ sequencing data but did not detect any donor carrying a clonal hematopoiesis-related mutation. Although previous study (Frick M et al. J Clin Oncol 2019;37:375-385) demonstrated that allogeneic hematopoietic stem cell transplantation from donors with clonal hematopoiesis seems safe, future studies are warranted to extend these results. We have included this content in discussion.

Point 2: Variant calling algorithm. Please provide details of the variant calling algorithm for chimerism measurement. Did you use integrative genomic viewer for chimerism? If it is the case, it is not an automated procedure that increases the chance of human error for chimerism measurement.

Response 2: Thank you for the correct comment. We used VarScan v.2.3.9 and Samtools mpileup to call variants and to count reads for chimerism calcultation and mutant allele burden, respectively. Integrative genomic viewer (IGV) was only used to check the presence of variants and strand bias. To clarify the point, we described the details in methods and Supplementary Figure 1. We absolutely agree that an automated procedure with minimal error should be developed to commercialize our algorithm. Through this study, we have successfully assessed the usability of simultaneous analysis algorithm of mutation and chimerism by manual. Now, we are developing an automated program to apply this algorithm for further validation study in a prospective manner with a large number of cases.

Point 3: Another pitfall of the conventional NGS methodology is the repetitive amplification of the same reads, which will affect the calculation of chimerism. In order to overcome this limitation, there is a lot of technology development incorporating bar-coded error-corrected sequencing methods such as smMIP probe development. I wonder if authors have a plan to apply this methodology to improve variant calling for allele frequency. At least it has to be included in the discussion. A very subtle change of chimerism would also affect significantly decision making for the clinical management of the case. Thus more accurate methodology for bar-coded error-corrected sequencing is needed for this purpose as well as more solid algorithm is required for variant calling.

Response 3: This is another good point. We totally agree with the reviewer’s opinion. Although we excluded low quality SNPs including low read depth, high background error rate and allelic imbalance to minimise the bias, there are still technical limitations such as repetitive amplification of the same reads. This can be overcome by barcoded error-corrected sequencing methods such as single molecule molecular inversion probes (smMIPs). In addition to automated program, we have a plan to apply adequate error-corrected sequencing method to improve variant calling for allele frequency. We have commented the limitation and necessity of error-corrected sequencing method along with two references (Genome Res. 2013;23: 843–854, Haematologica 2017;102:1549-1557).

Point 4:  If your goal of the study is to demonstrate that NGS based chimerism measurement is as useful as STR-based chimerism test, not only the correlation analysis between the two methods (as shown in Figure 1A) but also you need to calculate positive and negative predictive value for your test as well as sensitivity/specificity.

Response 4: Per your suggestion, we have calculated positive and negative predictive value for our test to diagnose complete donor chimerism. Previous studies suggested three criteria for complete chimerism including 99%, 95% and 90%. Thus, we calculated the values for each criteria. The overall agreements between the two methods were 100%, when we apply the 95% and 90% cut-off values for complete donor chimerism. When applying 99% cut-off value for complete chimerism, SNP-based NGS chimerism analysis showed 100% sensitivity, 91.7% specificity, 92.3% positive predictive value and 100% negative predictive value. Because our laboratory uses 95% cut-off value for complete chimerism, current agreement between two methods was 100% in qualitative comparison. We have described the overall data in the section 3.1.

Point 5: For section 3.2 (page 4, line 148), there is result from the 14 patients. Are there any predetermined criteria for case selection? For example, did you intend to include some of full donor chimerism and some of mixed chimerism as well as a case of donor cell-derived leukemia?

Response 5: As you mentioned, we have included all possible conditions after allo-HSCT including complete donor chimerism, mixed chimerism and donor cell-derived MDS to investigate the most effective approach for the simultaneous detection of the mutation and chimersim. We described the inclusion strategy in line 71-73.

Point 6: I was not sure the last case of donor cell-derived leukemia is a relevant case for this approach. If yes, please highlight what is the benefit of using simultaneous monitoring of chimerism and mutation based on NGS based method.

Response 6: Donor cell-derived hematologic malignancy is an infrequent complication after all-HSCT. Its diagnosis is difficult because blood cell count and chimerism analysis such as short tandem repeat (STR) are routine follow up tests. It can be delayed until blasts emerge in blood because cytopenia is not unusually observed in recipients and STR shows complete donor chimerism. Previous report reviewed literatures and identified more than 70 cases of donor cell leukemia. Time between allo-HSCT and occurrence of donor cell-derived leukemia (DCL) was various (median 30 months, range 1-279). Identification of donor cell origin was karyotype, fluorescence in situ hybridization, STR and sometimes adequate molecular analysis (Eur J Haematol 2018;101:570–574). Although donor cell origin of malignant cells can be verified by STR assay, it is important to clarify the genetic landscape and factors which contribute to DLC. It can help us to understand pathogenesis of DLC and to make better therapeutic plans. Some are associated with germline mutations such as CEBPA and DDX41 that might contribute to DLC development after gain a second somatic mutation (Leukemia 2017;31:1020–1022). We believe that DLC in our study is a relevant case to strengthen the usability of our simultaneous analysis.

We agree with the reviewer that we have not done a good job in highlighting the benefit and novelty of our analysis. Thanks to the reviewers’ comments, we have strengthened the benefit of simultaneous monitoring chimerism and mutation based on NGS based method.

Point 7:  Table 2 is labelled as Table 1 (line 176). Please add a column to label the case as mixed chimerism or donor-cell derived leukemia etc. That would help readers to have a better understanding of Table 2.

Response 7: We have changed the Table 1 to Table 2 and added a case type column in Table 2. Thank you for the suggestion.

Point 8:  Figure 3 does not include the definition of abbreviations used in the figure.

Response 8: We have included the definition of abbreviations used in the figure.

Point 9: Turn-around time of reporting has never been described in the paper.

Response 8: In the revised manuscript, we have described turn-around time of simultaneous analysis. It usually takes about 4 days (3-5 days). It takes similar time to mutation analysis using NGS because SNP-based chimerism calculation can be performed simply with minimal time (less than 30 minutes). We have commented turn-around time in discussion. Thank you for this comment.

Again, we appreciate very much the constructive and helpful comments. We did our best to be responsive to them. Thank you for taking the time and energy to help us improve the manuscript.

Reviewer 2 Report

The authors combined the detection of chimerisms and mutations to monitor MRD in14 patients who received an allo-HSCT during their treatment for MDS. 

The authors showed that data from STR and NGS were basically equivalent (at least from what I can see from Figure 3). The mutant allele burden was also detected via NGS and appears to have given the authors additional information to predict relapse. 

--> However, only three patients relapsed and could be further analysed. Of these most followed different disease progression which was evident by through the analysis. One patients relapsed with a doner derived mutation which was concluded from the combination of almost complete chimerism (99.8%) and the emergence of novel mutation in PHF6. I guess this is perhaps interesting but I'm unsure how useful this information actually is ... supposedly mutational analysis at the time of relapse would have supported the same conclusion ?

Given the small patient numbers and differences in the patient populations it is a bit unclear to me what the benefit of this combined approach is ? The authors should state more clearly when this approach would be used over others ? 

My feeling is that the dataset might be better suited as a case report rather than a research article. 

Author Response

Response to Reviewer 2 Comments

The authors combined the detection of chimerisms and mutations to monitor MRD in14 patients who received an allo-HSCT during their treatment for MDS.

First of all, we would like to thank you for the excellent comments that improved our manuscript after revision. We respond below in detail to each of the reviewer’s comments. In the following sections, you will find our responses to each of your points and suggestions.

Point 1: The authors showed that data from STR and NGS were basically equivalent (at least from what I can see from Figure 3). The mutant allele burden was also detected via NGS and appears to have given the authors additional information to predict relapse.

 Response 1: Yes, data from STR and NGS were basically equivalent especially in complete chimerism. There are increasing states of mixed chimerism, especially after reduced intensity conditioning regimens and after T-cell depletion. Mixed chimerism can have multiple meaning and clinical implications including disease relapse, graft failure or rejection. However, previous studies showed that mixed chimerism may remain stable over long time and be compatible with prolonged remission (Leukemia 2002;16:13-21). Because MRD monitoring is important to predict relapse after hematopoietic stem cell transplantation (HSCT) (Curr Opin Hematol 2018;25:425–432), it is valuable to monitor MRD and evolving mutations using NGS along with chimerism assay in those patients.

On the basis of your comments, we have added the significance of monitoring mutant allele burden by NGS assay after HSCT in Discussion. Thank you for this valuable comment.

Point 2: Only three patients relapsed and could be further analysed. Of these most followed different disease progression which was evident by through the analysis. One patients relapsed with a doner derived mutation which was concluded from the combination of almost complete chimerism (99.8%) and the emergence of novel mutation in PHF6. I guess this is perhaps interesting but I'm unsure how useful this information actually is ... supposedly mutational analysis at the time of relapse would have supported the same conclusion ?

 Response 2: Donor cell-derived hematologic malignancy is an infrequent complication after all-HSCT. Its diagnosis is difficult because blood cell count and chimerism analysis such as short tandem repeat (STR) are routine follow up tests. It can be delayed until blasts emerge in blood because cytopenia is not unusually observed in recipients and STR shows complete donor chimerism. Previous report reviewed literatures and identified more than 70 cases of donor cell leukemia. Time between allo-HSCT and occurrence of donor cell-derived leukemia (DCL) was various (median 30 months, range 1-279). Identification of donor cell origin was karyotype, fluorescence in situ hybridization, STR and sometimes adequate molecular analysis (Eur J Haematol 2018;101:570–574). Although donor cell origin of malignant cells can be verified by STR assay, it is important to clarify the genetic landscape and factors which contribute to DLC. It can help us to understand pathogenesis of DLC and to make better therapeutic plans. Some are associated with germline mutations such as CEBPA and DDX41 that might contribute to DLC development after gain a second somatic mutation (Leukemia 2017;31:1020–1022). Another important interesting condition is pre-existing clonal hematopoiesis in donor. Although previous study (Frick M et al. J Clin Oncol 2019;37:375-385) demonstrated that allo-HSCT from donors with clonal hematopoiesis seems safe, it is necessary to extend these results using comprehensive monitoring system. We believe that our simultaneous analysis using NGS is useful to evaluate various clinical conditions after allo-HSCT not only in MDS patients but also other hematologic disorders.

Point 3: Given the small patient numbers and differences in the patient populations it is a bit unclear to me what the benefit of this combined approach is? The authors should state more clearly when this approach would be used over others?

Response 3: We agree with you that we have not done a good job in highlighting the benefit and novelty of our analysis. Thanks to the reviewers’ comments, we have strengthened the benefit of simultaneous monitoring chimerism and mutation based on NGS based method. We appreciate very much the constructive and helpful comments of the reviewer in this regard.

Point 4: My feeling is that the dataset might be better suited as a case report rather than a research article.

Response 4: In this study, we developed a new analytic algorithm to monitor chimerism applicable to NGS multi-gene panel which was clinically used to detect mutation. To investigate the most effective approach, we have included all possible conditions after allo-HSCT including complete donor chimerism, mixed chimerism and donor cell-derived MDS (53 samples from 14 allo-HSCT). Then, we carefully evaluated the analytical and clinical performance of the new algorithm. Through this well-planned approach, we have successfully assessed the usability of simultaneous analysis algorithm of mutation and chimerism. You can think like that this manuscript includes interesting cases worthy of describing details. Now, we have highlighted the benefit of this combined analysis based on your recommendation. Please understand that we would like to present this study as an research article.

We would like to thank you again for a thoughtful review of the manuscript. We are confident that the new version of the manuscript will be greatly improved. We hope that you will find our responses to your comments satisfactory. Thank you for taking the time and energy to help us improve the manuscript.

Round 2

Reviewer 1 Report

All the queries requested are answered appropriately.